# Adipose-Derived Stem Cells for Facial Rejuvenation

**DOI:** 10.3390/jpm12010117

**Published:** 2022-01-16

**Authors:** Agnieszka Surowiecka, Jerzy Strużyna

**Affiliations:** 1East Center of Burns Treatment and Reconstructive Surgery, 77-069 Łęczna, Poland; jerzy.struzyna@gmail.com; 2Department of Plastic Surgery, Reconstructive Surgery and Burn Treatment, Medical University of Lublin, 20-093 Lublin, Poland

**Keywords:** facial rejuvenation, mesenchymal stem cells, liposuction, fat graft

## Abstract

The interest in regenerative medicine is increasing, and it is a dynamically developing branch of aesthetic surgery. Biocompatible and autologous-derived products such as platelet-rich plasma or adult mesenchymal stem cells are often used for aesthetic purposes. Their application originates from wound healing and orthopaedics. Adipose-derived stem cells are a powerful agent in skin rejuvenation. They secrete growth factors and anti-inflammatory cytokines, stimulate tissue regeneration by promoting the secretion of extracellular proteins and secrete antioxidants that neutralize free radicals. In an office procedure, without cell incubation and counting, the obtained product is stromal vascular fraction, which consists of not only stem cells but also other numerous active cells such as pericytes, preadipocytes, immune cells, and extra-cellular matrix. Adipose-derived stem cells, when injected into dermis, improved skin density and overall skin appearance, and increased skin hydration and number of capillary vessels. The main limitation of mesenchymal stem cell transfers is the survival of the graft. The final outcomes are dependent on many factors, including the age of the patient, technique of fat tissue harvesting, technique of lipoaspirate preparation, and technique of fat graft injection. It is very difficult to compare available studies because of the differences and multitude of techniques used. Fat harvesting is associated with potentially life-threatening complications, such as massive bleeding, embolism, or clots. However, most of the side effects are mild and transient: primarily hematomas, oedema, and mild pain. Mesenchymal stem cells that do not proliferate when injected into dermis promote neoangiogenesis, that is why respectful caution should be taken in the case of oncologic patients. A longer clinical observation on a higher number of participants should be performed to develop reliable indications and guidelines for transferring ADSCs.

## 1. Introduction

The process of healing has fascinated people for centuries. One of the first descriptions of the human body’s regenerative abilities originates from ancient Greece. In the myth, the titan Prometheus was chained to a rock, and every day an eagle ate a part of his liver, which would regenerate and grow back again. Nowadays, regenerative medicine is a dynamically developing branch of aesthetic surgery. 

The term ‘regenerative medicine’ was defined by Daar and Greenwood in 2007 as a multidisciplinary branch of medicine, which stimulates the human body to repair and heal malfunctioning tissues [1]. In aesthetic medicine, the term ‘biostimulation’ is often used to describe the impact of injectables on the function of dermal fibroblasts [2]. Even though more than a decade has passed since a unified description of regenerative medicine has been established, there are still some semantic aspects that need to be clarified. This is especially noticeable with regard to autologous-derived agents and a wide range of different products and nomenclature that varies between authors. However, the aim of this paper was not an attempt to establish nomenclature, but rather to describe the potential use of autologous-derived agents in facial rejuvenation.

## 2. Harvesting Autologous-Derived Agents 

A wide range of autologous products are used for the purpose of facial rejuvenation. The most commonly applied therapies include agents collected from peripheral blood, such as platelet-rich plasma (PRP) or fibrin. These agents are easy to collect and require only cannulation of a peripheral vein. Adipose-derived products are more demanding to harvest and require surgical experience. Fat tissue is a source of micronized fat graft (Microfat) or homogenized gel form (Nanofat) and stromal vascular fraction (SVF). PRP action depends on stress-activated platelets that secrete growth factors and proteins [3] and not on stem cells as is wrongly advertised by some providers. α- granules contained in platelets can secrete over 4000 different proteins. By stimulating fibroblasts, they induce the secretion of collagen, elastin, and the self-synthesis of hyaluronic acid. They also release growth factors for fibroblasts (FGF—fibroblast growth factor), PDGFs (platelet-derived growth factor), TGFβ (transforming growth factor), EGF (epidermal growth factor), and VEGF (vascular endothelial growth factor) [4,5,6,7,8,9]. The main indications for PRP in aesthetics are rejuvenation and fine line correction.

The primary sources of stem cells include bone marrow [10] and subcutaneous fat tissue [11]. Subcutaneous fat tissue contains 500 times more stem cells than bone marrow [11]. Several issues are subject to discussion concerning subcutaneous fat tissue harvesting techniques (Table 1). Subcutaneous fat tissue is feasible for collection in most patients. However, there are some data indicating that certain body areas are richer in stem cells. Padoin et al., as well as Tsekouras, suggested that lipoaspirates from the lower abdomen and the inner thigh had higher concentrations of adipose-derived stem cells (ADSCs) [12,13]. Di Taranto divided subcutaneous white adipose tissue (SWAT) into two parts: superficial and deep. ADSCs harvested from the superficial parts of the SWAT had better survival and presented higher expression of vascular endothelial growth factors (VEGFs) [14]. However, harvesting subcutaneous fat tissue from superficial parts brings the risk of irregularities. The Coleman procedure was first introduced in 1987 [15]. He used local anaesthesia with lidocaine and epinephrine and collected fat with a 2 mm diameter blunt cannula with a suction value amounting to 1 mL on a 10 mL syringe. He believed that too much suction would destroy the cells. There is data suggesting that negative pressure and donor site do not influence ADSC survival for culturing and yield preparation [16], but in terms of clinics and facial lipofilling, there might be a higher resolution of the transfer harvested traumatically [15,17,18,19]. For assisted liposuctions, no differences were observed between power-assisted and ultrasound-assisted liposuction [20], but laser-assisted procedures performed worse than power-assisted [21]. Fontes et al., in a precise review of fat harvesting techniques, described the impact of the cannula—its diameter as well as hole number and diameter—on fat graft survival. They concluded that the size of the cannula should provide minimal shear stress to avoid cell breakage [22]. 

Several techniques of fat tissue processing have been described. The most popular among them are procedures that involve centrifugation. The parameters of spinning differ between the authors and sets [15,23,24,25,26]. Gravity separation and sedimentation requires no specific equipment [32]. There are some sets that enable the washing and infiltration of the fat graft [27]. Collecting ADSC from the fat graft requires micronization of the fat particles [26,28,29,30,31]. In the Arthrex procedure, the lipoaspirate is spun in ACP double syringes for four minutes at 2500 RPM in a Horizon 24-AH swing-out rotor centrifuge (Drucker Diagnostics LLC, Port Matilda, PA, USA). Subsequently, the oil from the destroyed adipocytes on top and the aqueous fraction, containing the anaesthetic and blood cells, are removed. In another step, the aspirate is fractionated by 30 rapid passages through a 1.2 mm female connector and spun for a second time for 4 min at 2500 RPM [26]. The final product that is obtained is the stromal vascular fraction (SVF). Stromal vascular fraction (SVF) contains significantly fewer adipocytes than the fat graft, while the volume of the fat graft comes in 90% from adipocytes [33]. Both SVF and the fat graft consist of numerous active cells such as pericytes, preadipocytes, immune cells, and extra-cellular matrix [33]. That is why most surgeons do not transfer stem cells in an office procedure for aesthetic purposes when the procedure is conducted according to the FDA principles. The proper term for this product is stromal vascular fraction, whereas stem cell transfers require cell incubation and counting [34]. 

## 3. The Powerful Action of the ADSCs in Facial Rejuvenation

SVF is a source of stem cells, but its regenerative properties also depend on other active cells (Table 2). ADSCs are multipotent cells, however, in vivo they do not proliferate but act as regulatory cells instead [35]. They are characterized by the expression of CD34^+^, CD44^+^, CD31^−^, and CD45^−^ on the surface [33]. They have a high proliferation ability, and they promote angiogenesis in the dermis [33,36]. They also secrete many growth factors, among others for fibroblasts, the endothelium, as well as anti-inflammatory cytokines. ADSCs stimulate tissue regeneration by promoting the secretion of extracellular proteins, such as collagen and elastin, but also metalloproteinases [37]. The mechanisms triggered by ADSCs are nonspecific paths of immune response [35]. In the paracrine mechanism, they secrete exosome and microvesicles containing proteins, nucleic acids, lipids, and enzymes [38]. Most authors recognize the longevity of the results of the transfer as the actual survival time of ADSCs, but in aesthetic patients the exact in vivo survival time of cells is difficult to determine. Zhang et al., in a mouse model, achieved 28-day survival of cultured stem cells injected intradermally [39]. Another cell population responsible for a strong regenerative attribute of SVF are pericytes. They are also multipotent cells that do not proliferate in vivo into adipocytes or fibroblasts. Their role is limited to neovascularization [40]. Adipocytes are highly active cells with paracrine and autocrine abilities [41]. SVF also contains the extracellular matrix [33] and immune cells that are responsible for the cellular paracrine dialogue [42]. Some authors use sieves to clarify the SVF gel [43], but in our opinion this technique might impoverish the product. 

SVFs stimulate collagen I, II, III, and V, oxytalan, and elastin secretion [36,44]. Restoration of ECM fibres prevents skin laxity and atrophy [36]. Activated fibroblasts, via different pathways Wnt/β-catenin, PI3K/Akt, insulin-like growth factor (IGF), IL-1, and TNF-α, secrete extracellular matrix proteins, mainly collagen type I and III [44]. In a micropig model, ADSCs were injected intradermally and skin density was evaluated after a month. Western blot showed an increased expression of dermal collagen [45].

ADSCs improve lesions caused by photoaging. They secrete antioxidants and cytokines that neutralize the effects of the primary indicators of skin damage: UVB and reactive oxygen species (ROS) [38,44]. ROS promote inflammation, damage to cellular membrane, as well as alterations in DNA, RNA, and proteins of the extracellular matrix (ECM). ADSCs secrete growth factors, including the hepatocyte growth factor (HGF) or VEGF, that protect cells from oxidative cells. Interleukin-6 (IL-6) reduces oxidative stress via promotion of the activator of transcription 3 (STAT3), nuclear factor erythroid 2-related factor 2 (Nrf2), and superoxidative dismutase (SOD). Nrf2 downregulates NOX1 and NOX4 responsible for lipid peroxidation. They also secrete glutathione peroxidase (GPx), SOD, and catalase to upregulate antioxidant response, as well as inhibit secretion of myeloperoxidase (MPO) [39,44]. In a rat model, stem cells suppressed the activity of free radicals on the DNA. They inhibit hypoxia and apoptosis by upregulating BCl-2 [38,44]. UVB irradiation is disactivated by the suppression of mitogen-activated protein kinases (MAPKs) and nuclear factor kappa B NF-κB [38].

Photoaging is also manifested by hyperpigmentation. These lesions are acquired pigmentary lesions that mainly affect women. Risk factors include exposure to UV radiation, hormonal changes, and genetic predisposition [50]. These are areas of increased melanin concentration in the skin and epidermis. The accumulation of UV radiation causes inflammation while also stimulating melanogenesis and angiogenesis. Hyperpigmentation is caused not only by the accumulation of melanin, but also occurs due to pathologically dilated vessels [46]. Adipose-derived stem cells secrete TGF-β1, which is a suppressor of tyrosinase, an enzyme necessary for melanin synthesis [44]. 

Charles-se-Sá et al. transferred isolated stem cells (CD105^+^/CD90^+^/CD73^+^/CD146^+^/ CD14/CD45^−^/CD34^−^) and assessed histopathological specimens of skin from the injection site. Skin biopsies taken before the treatment were diagnosed with elastosis due to the degradation and disorganization of elastic fibres, as well as mild mononuclear dermis infiltration. After the SVFs, the dermis was richer in oxytalan elastic fibres, and a decrease in elastosis was observed in the reticular dermis. The dermis showed new capillary vessels, and the skin was more hydrated [47]. Stem cells inhibit metalloproteinases MMP-1, MMP-2, MMP-3, MMP-9, and MMP-13 [48]. 

The most recent studies report the presence of ADSCs in subcutaneous tissue, which might suggest their ultimate role in skin repair and wound healing [49]. During injury, expression of the CXCR-4 molecule on ADSCs’ surface increases, and stem cells are recruited to modulate local inflammatory response. ADSCs have the potential to migrate to the site of injury. It has been proven that overexpression of SDF-1 protein is responsible for cell migration. In the initial phases of wound healing, they promote a shift between macrophage population from M1 to M2 and stimulate secretion of anti-inflammatory TNF and IL-10. In later phases, ADSCs stimulate angiogenesis via the secretion of growth factors including VEGF, PDGF, IGF, HGF, b-FGF, SDF-1, TGF-β, and GDF11. During the proliferation phase, ADSCs promote the secretion of ECM proteins [49], as well as secrete fibroblast chemokines [45]. 

## 4. Clinical Indications for SVF

The SVF gel can be transferred separately or with the fat transfer (Table 3). CAL (cell-assisted lipotransfer) is a simultaneous transfer of these two adipose-derived agents [51,52]. It was first introduced in 2008 by Yoshimura [53]. The primary reason for using a combination of this kind is the unpredictable survival time of fat grafts. Depending on the author and technique, one-year graft survival varies from 10% to 90% [50]. Adding SVF over the fat graft stimulates the process of neovascularization [54,55], as in the early post-transplant period, fat molecules are nourished by osmosis alone [55]. Due to Yoshimura’s rule, fat particles smaller than 200 µm are the least vulnerable to apoptosis [56]. The effects of the CAL are shown in Figure 1.

Adding SVF over the lipotransfer is one of the factors prolonging the result. Schnedel used a 3D computer volumetric analysis to evaluate long-term results. In a 12.6-month observation, 68% of the transferred graft survived [57]. Similar observations were done by Yating Tin et al. The survival time of CAL was higher than in the case of fat transfer alone (77.6% vs. 56.2%) [58]. Adding SVFs to fat grafts may prevent the necessity of secondary procedures and reduce the cost of the treatment [59]. 

Fat grafts enriched with SVFs have a potential antifibrotic effect. Almadori et al., treated 62 patients with oro-facial fibrosis in systemic sclerosis. Lipotransfer and SVFs reduce fibrosis due to suppression of fibroblast proliferation. The clinical outcomes, including mouth function, were improved in the observation. The levels of transforming growth factor (TGF-β1) and connective tissue growth factor were reduced in the treated population [43].

CAL can be conducted simultaneously with other surgical procedures, such as transconjunctival lower blepharoplasty [60], facelifting [44], threads, or to promote wound healing [61,74].

Jiang et al. described the surgical technique for lower eyelid blepharoplasty and simultaneous SVF gel injection. They used a Coleman procedure for fat harvesting. A transconjunctival fat pad removal was performed using an incision 5 mm below conjunctiva, the opening of the capsulopalpebral fascia was done typically, protruding periorbital fat was resected, and no sutures were left on the conjunctiva. After upright positioning of the patients, SVF gel was injected using one to two entry points with a 0.9mm blunt cannula to position the fat graft between the orbicularis oculi muscle and the infraorbital septum [43].

Berbardini et al. reported successful combined protocols for aesthetic surgeries and simultaneous fat injections. The surgery started from aesthetic surgeries: minimal incision vertical endoscopic lift (n = 51), primary blepharoplasty (n = 35), neck lift (n = 23), and revisional blepharoplasty (n = 12), which were performed in a typical way. In the second step during the same procedure, the harvested and processed fat was injected in superficial layers with a 23G sharp needle. The aim of fat transfer was to restore volume and improve skin quality [62].

Simultaneous fat grafting and PDO barbed threads were reported by Surowiecka [63]. The procedure started with fat injections following the ACA technique [64] and was followed by placing in the subcutis layer a total number of six barbed threads from four different entry points to elevate the midface and to improve the jawline. One entry point was done over the zygomatic ligament, the first thread was directed to the nasolabial folds, through masseteric ligament, placing the end of the threads just behind the nasolabial fold. The second entry point was done over the preauricularis ligament and two threads with a fanning technique were directed to the mental area [63]. 

Transferring SVFs alone improves skin density and thickness as well as dermis density [47,65]. The results were stable in a 12-month observation [66,67]. The improvement of skin density provides the secondary effect of volumization and tissue elevation [38,57,58]. In a comparison of nanofat, microfat, and SVF, the count of viable cells in SVF increased moderately in comparison with nanofat and microfat [43]. The results are shown in Figure 2. 

Mesenchymal stem cells undergo senescent changes [68], as does the fat tissue. In in vitro observation, stem cells from elder donors are characterized by an elevated expression of the p53 gene, b-galactosidase, and a decrease in intrinsic antioxidant mechanisms, becoming more vulnerable to apoptosis [68]. Aging cells are found during the G1-phase, and their apoptosis is induced by b-galactosidase and p53 [44,75]. Stem cells may influence senescent lesions in other cells [75]. With age, the number of fat tissue M2 macrophages increases, whereas the number of M1 macrophages decreases. M2 macrophages are known to secrete pro-inflammatory cytokines responsible for atherosclerosis. The macrophages shift decreases the ability to neutralize free fatty acids and promotes weight gain. With time, the ability to proliferate and secrete adiponectin also decreases [76]. Fat tissue aging may impair the final results of lipotransfer in elderly patients even with SVFs. 

SVF can be successfully combined with lasers in a combined therapy of scars. Fat grafting has been reported in burn scars in both adults and children. The indications for the lipofilling of post-burn scars include hypotrophic scars without improvement after pressure garments and fibrotic scars around joints including tendon adhesions [69,70]. ADSC were reported to efficiently treat post-ablative laser wounds even in skin type Fitzpatrick III-IV [71]. They significantly reduced erythema after carbon dioxide laser and melanin treatment. The increase of collagen in the skin treated with ADSC was not statistically significant, but there was a short observation time. A topical use of cultured ADSC with niacinamide was reported by Lee. The protocol started from ablative laser CO2 resurfacing. Topical ADSC with niacinamide improved wrinkles and skin texture, as well as skin pigmentation [72]. Verpaele combined liquid nanofat with microneedling and gained satisfactory results [73]. Considering these results, SVFs could also be combined with microneedling during a procedure. 

## 5. Complications

Complications can be related to liposuction and to fat injections. The total rate of reported complications after liposuction is 5% [77] For SVF purposes, a small amount of fat tissue is required; however, the risk associated with the procedure remains. The most common side effects of fat tissue harvesting are mild and transient: primarily hematomas, oedema, and mild pain [78]. A total of 23% cases of reported deaths were caused by pulmonary thromboembolism [77]. Other serious complications include bleeding [79] and fat embolism [77]. Cases of abdominal visceral damage have been described as well [77].

As far as the fat graft is concerned, several cases of graft migration, formation of calcifications and cysts, as well as ischemia, necrosis, and vision loss have been described [80]. Most often these side effects are transient and minor [26]: primarily oedema and hematomas. As a facial soft tissue filler, fat can cause vascular occlusion, which is why a decent anatomy assessment and using blunt cannulas is recommended [26].

ADSCs do not proliferate in vivo and do not undergo neoplastic transformation per se [35], but they promote neoangiogenesis. There is a risk of promoting the neoplastic vascular network, local recurrence, and metastases [80,81]. There are no guidelines for autologous cell therapy in oncologic patients. ADSCs stimulate the process of angiogenesis, therefore the risk of stimulating the neoplastic vascular network and promoting metastasis seems to be very high [82]. One study revealed an improvement in healing and a longer period of remission in head and neck squamous cell carcinoma after the defects of soft tissues had been filled with autologous agents [83]. On the other hand, there are studies showing local progression of ductal adenocarcinoma in the breast and accelerated formation of distant metastases [84]. Klinger et al. in a multicentre study reported that fat transfer for breast reconstruction in breast cancer patients did not worsen the outcome. They used the Coleman technique, without enriching the grafts with adipose-derived stem cells [85]. Calabrese et al. transplanted fat grafts enriched in SVF. It was a prospective study with 41 patients with breast cancer G1 enrolled. Patients diagnosed with breast cancer G2 were treated with lipotransfer alone, and G3 were a control group. Adding SVF to the fat graft in the G1 group did not increase oncological recurrence [86]. In aesthetic indications, it is extremely important to take a proper medical history and examination of the patient before the procedure. 

### Future Trends

The main limitations of lipotransfers and SVF transfers are the unpredictable outcomes and harvesting procedure. Complications after liposuction can be life-threatening. That is why novel approaches in tissue harvesting are being developed. One of the ideas is to store lipoaspirate in a tissue bank to use it later. Cryopreservation proved to be successful, and the storing process did not influence the number or viability of mesenchymal stem cells [87]. Ohashi et al. divided the lipoaspirate into two parts. The first was injected during the same day of the procedure and the second part was centrifuged with Ringer’s solution, washed, added with cryoprotective solution, then transferred into liquid nitrogen and stored at −196 °C [87]. 

Fat tissue is highly immunogenic, so allogenic fat transfers are extremely difficult to obtain. There is a case of a successful allogenic fat transfer into post-radiative ulcerations due to cutaneous T-cell lymphoma. The transfer was possible because of the chimeric state between the donor and recipient, who were brothers [88].

Some investigators work on different scaffolds and growth factors that could be safely added to fat grafts to improve its survival [80]. The aim of adding biomaterials is to avoid the necrosis of the stem cells and improve revascularization. There are few scaffolds based on hyaluronic acid reported, as well as beta-glucan or poly(lactide-co-glycoside) [80].

Finally, there are observations that simple techniques, such as botulin, can have a beneficial influence on fat grafting results. By relaxing the moving parts of the face before transferring fat, the mechanical stress of the muscles is diminished, and fat particles are believed to last longer [35].

## 6. Conclusions

Regenerative medicine is still developing. The mechanism of action of ADSCs and their multidirectional impact on dermis rejuvenation fascinates physicians. ADSCs secrete antioxidative agents, as well as promote wound healing and ECM protein secretion. There are several reports proving that ADSC injections improve skin density and have anti-wrinkle properties. Skin hydration improves and there is a global improvement in appearance. There are some limitations, however. First, it is very difficult to compare studies because of the differences in techniques used. Most fat and stem cell transfers performed due to aesthetic indications are in-office, one-stage procedures without cytometry and cell counting. Stem cell harvesting systems differ and surgeons use different techniques. It seems that the survival and vitality of stem cells is strongly associated with the harvesting technique, and there are no unequivocal guidelines in this area either. Most of the studies provide only a one-year or two-year follow-up on a small number of volunteers. A longer clinical observation on a higher number of participants should be performed to develop reliable indications and guidelines for transferring ADSCs. 

## 7. Key Points

ADSCs are efficient and safe for facial rejuvenation but harvesting ADSC requires some technical skills and adequate equipment. 

They improve skin quality in many mechanisms, which makes them superior to other known anti-aging agents. They promote the remodelling of the dermis, which is important in scar treatment and improvement of dermis thickness. ADSCs have an antioxidative effect and can protect fibroblast DNA from free radicals. Oxidative stress is perpetual due to UV radiation, pollution, diet, and smoking. Furthermore, they stimulate secretion of collagen, elastin, and anti-inflammatory cytokines. These features make ADSC an ideal and natural anti-aging solution.

## Figures and Tables

**Figure 1 jpm-12-00117-f001:**
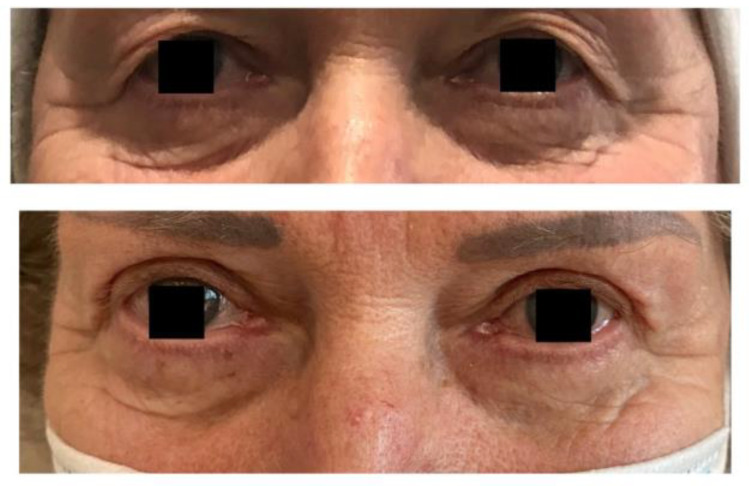
Results of CAL and upper blepharoplasty.

**Figure 2 jpm-12-00117-f002:**
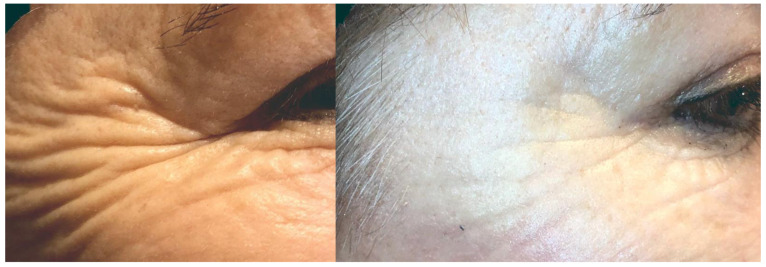
Results of skin improvements 6 months after SVF injection.

**Table 1 jpm-12-00117-t001:** Clinical studies and reviews considering usage of autologous agents in rejuvenation.

Study	Study Type	Patients and Methods	Outcomes	Conclusion
Adipose-Derived Mesenchymal Stem Cells, Platelet-Rich Plasma and Biomaterials as New Regenerative Strategies in Chronic Skin Wounds and Soft Tissue Defects [3]	Review	72 articles met the criteria	84% of studies showed effective outcomes of autologous therapies	PRP, ADSC are safe and can be used for the therapy skin defects and wounds.
The in vitro effect of different PRP concentrations on osteoblasts and fibroblasts [4]	Experimental	Peripheral blood was collected from three healthy volunteers, human oral fibroblasts and osteoblasts were cultured with activated and non-activated PRP as various concentrations	PRP stimulates fibroblasts and osteoblasts to proliferate. The maximum effect was obtained at platelet concentration of 2.5×.	Platelet concentration in PRP may affect the final results. Study showed that platelet concentration of 2.5× is most optimal.
Fibroblastic response to treatment with different preparations rich in growth factors [5]	Experimental	Sixteen fibroblast cultures obtained from three different anatomical sites (skin, synovium and tendon) of 16 donors	Maximum proliferation rate of fibroblasts was obtained with PRP with twofold or fourfold platelet concentration. PRR stimulated HA synthesis (*p* < 0.05).	Platelet concentration in PRP may affect the final results.
A systematic review of the safety and effectiveness of platelet-rich plasma (PRP) for skin aging [6]	Review	24 studies, 480 patients after PRP	High satisfaction was noted among patients even though only 50% of skin improvement was observed.	PRP injections are safe and beneficial in skin aging.
Sources of processed lipoaspirate cells: influence of donor site on cell concentration [12]	A prospective cross-sectional study	25 females who underwent liposuction in four or more zones.	The lower abdomen and the inner thigh may have higher processed lipoaspirate cell concentrations.	Different body areas contain various number of active cells, which might influence the final outcome of lipotransfer.
Comparison of the viability and yield of adipose-derived stem cells (ASCs) from different donor areas [13]	A prospective study	40 females who underwent liposuction. Lipoaspirates and SVF were processed.	Inner and outer thigh showed a significantly higher number of ADSC compared to abdominal, waist, and inner knee samples (*p* < 0.05).	Different body areas contain various number of active cells, which might influence the final outcome of lipotransfer.
Qualitative and quantitative differences of adipose-derived stromal cells from superficial and deep subcutaneous lipoaspirates: a matter of fat [14]	A prospective study	16 females who underwent liposuction for elective breast augmentation. Three cadavers to collect full-thickness skin and abdominal wall specimens	Superficial adipose tissue contained a higher stromal tissue compound, along with a higher proportion of CD105-positive cells, compared with deep adipose tissue.	Layers of fat tissue contain various number of active cells, including stem cells which might influence the final outcome of lipotransfer
The Influence of Negative Pressure and of the Harvesting Site on the Characteristics of Human Adipose Tissue-Derived Stromal Cells from Lipoaspirates [16]	A comparative study	15 healthy volunteers, who underwent tumescent liposuction. ADSC were isolated and cultured.	Higher initial cell yields from the outer thigh region than from the abdomen region. Negative pressure did not influence the cell yields from the outer thigh region, whereas the yields from the abdomen region were higher under high negative pressure than under low negative pressure.	For in vitro culturing and for use in tissue engineering, negative pressure while harvesting lipoaspirates does not influence the outcomes.
Optimizing harvesting for facial lipografting with a new photochemical stimulation concept: One STEP technique™ [17]	A prospective study.	245 patients who underwent facial lipofilling	The novel technique of STEP™ showed good results in facial lipofilling.	Long-term results of skin improvement were seen starting from 2 month.
Influence of negative pressure when harvesting adipose tissue on cell yield of the stromal vascular fraction [18]	A comparative study	3 patients, 6 different harvesting techniques. Cell yielding was performed.	Negative pressure is a factor influencing the number of SVF cells harvested	Harvesting techniques may influence the number of stem cells in lipoaspirate.
Low harvest pressure enhances autologous fat graft viability [19]	A comparative study	3 patients who underwent lipoaspiration at two pressures: high −760 mmHg and low −250mmHg. Cell counting.	The cell count was 47% higher when tissue was aspirated with low pressure.	Harvesting techniques may influence the number of stem cells in lipoaspirate.
Ultrasound-assisted liposuction does not compromise the regenerative potential of adipose-derived stem cells [20]	A prospective study	3 females who underwent ultrasound-assisted liposuction (UAL), cell culturing (CD34+/CD31−/CD45−) and mice injecting	Cutaneous regeneration and neovascularization were significantly enhanced in mice treated with ADSC harvested with UAL	UAL is a successful method of obtaining fully functional ADSC for regenerative medicine purposes.
Isolation of human adipose-derived stromal cells using laser-assisted liposuction and their therapeutic potential in regenerative medicine [21]	A prospective study	12 females who underwent laser-assisted liposuction (Nd-YAG), cell culturing (CD34+/CD31−/CD45−) and mice injecting	Laser-assisted liposuction appears to negatively impact the biology of ASCs	Cell harvest using suction-assisted liposuction is preferable.
Clinical treatment of radiotherapy tissue damage by lipoaspirate transplant: a healing process mediated by adipose-derived adult stem cells [23]	A prospective study	20 patients with radiation damage, 31 months follow-up.	Transplanted tissue stimulated neoangiogenesis and tissue regeneration.	SVF is a safe and useful method to treat radiation damages.
Effects of a new centrifugation method on adipose cell viability for autologous fat grafting [24]	A prospective study	10 patients underwent Coleman technique; fat aspired from 10 patients was centrifuged at 1300 rpm for 5 min and from 10 patients at 3000 rpm for 3 min	1300 rpm resulted in better density of adipose tissue, with good cell viability and increased ability to preserve a significant number of progenitor cells	Forces of centrifugation may impact the final results of lipofilling.
Comparison of fat maintenance in the face with centrifuge versus filtered and washed fat [25]	A prospective single-blind analysis	32 healthy patients undergoing nasolabial fold fat transplantation	No significant difference in the survival of grafted fat between the fat-processing with centrifuge at 3400 rpm for 1-min and fat washed in the sieve	Centrifugation does not decrease the survival of cells in fat graft.
Stromal vascular fraction and emulsified fat as regenerative tools in rejuvenation of the lower eyelid area [26]	An observative study	16 patients underwent tumescent liposuction and injection of SVF and emulsified fat into the lower eyelid area.	Clinical outcomes were rated as exceptional, very improved, or improved in all patients, with an average GAIS score of 1.6. No serious adverse events occurred.	The study’s results suggest that SVF and emulsified fat are safe and effective tools for skin rejuvenation and correction of volume deficiencies in the lower eyelid area.
Centrifugation versus PureGraft for fat grafting to the breast after breast-conserving therapy [27]	A prospective study	30 patients who received fat grafts into breasts either after centrifugation or by washing. 30 months follow-up. BREAST-Q analysis.	No significant difference in BREAST-Q between two groups.	Macrofat grafting is of an unpredictable duration.
Conventional vs. micro-fat harvesting: how fat harvesting technique affects tissue-engineering approaches using adipose tissue-derived stem/stromal cells [28]	A prospective study	10 patients, one side harvested with a conventional fat harvesting by the Coleman cannula (3 mm, one-hole blunt tip) and the micro-fat-harvesting technique by the st’RIM cannula (2 mm, multi-perforated hole blunt tip) on contralateral area.	Viability and migration of isolated ADSC obtained from micro-harvested lipoaspirates were significantly higher.	The different sizes and surface of fatty tissue obtained by using different cannula sizes influence the effects.
Nanofat grafting: basic research and clinical applications [29]	A prospective study	67 patients underwent facial lipofilling with a 27-gauge needle.	Adipose-derived stem cells were richly present in the nanofat sample	Nanofat is a source of ADSC.
The efficacy of autologous Nanofat Injections in the treatment of infraorbital dark colouration [30]	A prospective study	10 female patients, emulsified fat tissue injection into the lower eyelid for dark circles.	Significant improvement or improvement was seen in 70% of cases.	Emulsified fat is a source of regenerative cells.
Correction of Dark Coloration of the Lower Eyelid Skin with Nanofat Grafting [31]	A prospective study	19 patients underwent simultaneous transconjunctival lower eyelid blepharoplasty with nanofat injection into lower eyelid	The procedure was safe, and all patients had improvement.	SVF can be simultaneously injected with lower eyelid blepharoplasty.

**Table 2 jpm-12-00117-t002:** Role of ADSC in skin rejuvenation.

Study	Study Type	Patients and Methods	Outcomes
Adipose-derived cellular and cell-derived regenerative therapies in dermatology and aesthetic rejuvenation [36]	Review	Review summarizes the use of adipose-derived products in hair growth, scar improvement, skin ischemia-reperfusion recovery, and facial rejuvenation	Cellular and cell-derived products are safe and effective in skin rejuvenation
Anti-Aging Effect of Adipose-Derived Stem Cells in a Mouse Model of Skin Aging Induced by D-Galactose [39]	Experimental	Six-week-old nude mice were subcutaneously injected with D-gal daily for 8 weeks	Transplanted ADSC were detectable for 14 days. ADSC inhibited advanced glycation, increased the SOD level and decreased the malondialdehyde level.
Pericytes and their potential in regenerative medicine across species [40]	Review	92 articles on pericytes. The regenerative potential of human pericytes (CD146+/CD45−/CD34−) was evaluated	Human pericytes have a regenerative potential, stimulate neoangiogenesis.
Comparison of the Efficacy and Safety of Cell-Assisted Lipotransfer and Platelet-Rich Plasma Assisted Lipotransfer: What Should We Expect from a Systematic Review with Meta-Analysis? [44]	Review	Evaluation of the efficacy and safety of CAL and PRP, 36 studies, 1697 patients	CAL and PRP-assisted lipotransfer significantly improved the fat survival rate (CAL vs. non-CAL: 71% vs. 48%, *p* < 0.0001; PRP vs. non-PRP: 70% vs. 40%, *p* < 0.0001; CAL vs. PRP: 71% vs. 70%, *p* = 0.7175
Adipose-derived stem cells and their secretory factors as a promising therapy for skin aging. [45]	A prospective study	3 micropigs, ADSCs injected intradermally, twice in a 14-day interval	ADSCs and their secretory factors can be used in cosmetic dermatology and anti-aging medicine.
The vascular characteristics of melasma [46]	A prospective study	50 Korean women with melasma, Immunohistochemistry to determine the expression of factor VIIIa-related antigen and VEGF in melasma.	The expression of VEGF was significantly increased in melasma
Mesenchymal Stem Cells from Adipose Tissue in Clinical Applications for Dermatological Indications and Skin Aging [47]	Review	248 articles regarding skin aging	ADSC have a beneficial impact on skin aging, however further studies are necessary to establish the optimal, long-lasting and, importantly, safe strategies for ADSCs.
Application of adipose-derived stem cells in photoaging: basic science and literature review [48]	Review	178 articles regarding photoaging and ADSC	ADSCs are potential to address photoaging problem and might treat skin cancer.
Hopes and Limits of Adipose-Derived Stem Cells (ADSCs) and Mesenchymal Stem Cells (MSCs) in Wound Healing [49]	Review	159 articles regarding ADSC in wound healing	ADSC can be used as a therapeutic strategy in wound healing and skin aging.
Innate Immune Control of Adipose Tissue Homeostasis [42]	Review	123 articles	Adipose immune cells play a crucial role in maintaining local homeostasis and contributes to the regulation of systemic metabolism

**Table 3 jpm-12-00117-t003:** ADSC clinical application.

Study	Study Type	Patients and Methods	Outcomes
Subcutaneous Injections of Nanofat Adipose-derived Stem Cell Grafting in Facial Rejuvenation [51]	Prospective study	50 patients for non-surgical facial rejuvenation were enrolled. They underwent subcutaneous nanofat injections. All patients confirmed an improvement in skin quality and a lifting effect.	Nanofat is a safe and efficient method in dermal rejuvenation.
Cell-assisted lipotransfer (CAL) for cosmetic breast augmentation-supportive use of adipose-derived stem/stromal cells [53].	Prospective study	70 patients (60 breast augmentation), rest face rejuvenation. The total volume of harvested fat was 1118, while CAL volume to the left breast was 268 mL and 277 to the right.	Postoperative atrophy of injected fat was minimal. Cyst formation or microcalcification was detected in four patients. Almost all the patients were satisfied with augmentation.
Platelet Rich Plasma (PRP) Improves Fat Grafting Outcomes [54]	An original study	A description of a method of PRP and microfat (Coleman technique) combination and usage for facial rejuvenation.	Addition of PRP to fat grafts offers a better fat grafting survival, a less bruising and inflammation reaction, and easier application of fat grafts due to liquefaction effect of PRP.
Nanofat-derived stem cells with platelet-rich fibrin improve facial contour remodelling and skin rejuvenation after autologous structural fat transplantation [55]	A comparative study	62 patients with soft tissue depression or signs of aging who underwent combined nanofat, PRF, and autologous fat structural transplantation had been compared to 77 control group patients who underwent traditional autologous fat transplantation. Flow cytometry after one of the following rabbit anti-human primary antibodies was added: CD29-PE, CD44-PE, CD49d-PE, CD54-PE, CD90-PE, or CD105-PE. Incubation with CD34-PE, CD45-PE, CD106-PE. Microfat, nanofat, and PRF were mixed. Follow-ups occurred 7 days, 3 months, 6 months, and 12 months after operation	Transplants that combine newly isolated nanofat, which has a rich stromal vascular fraction (SVF), with PRF and autologous structural fat granules may therefore be a safe, highly effective, and long-lasting method for remodelling facial contours and rejuvenating the skin.
The fate of adipocytes after non-vascularized fat grafting: evidence of early death and replacement of adipocytes [56]	Experimental	Cultured human adipocytes and cellular components of fat tissue	Adipocytes are viable for 3 days, and those located within 300 μm of the tissue edge survived.
Enriched autologous facial fat grafts in aesthetic surgery: 3D volumetric results [57]	A prospective study	12 females, 50 cc of autologous lipoaspirated fat, SVF injection (mean 18,4 cc). Evaluation of the results with 3dMD photogrammetric system, average follow up 12.6 months	Graft survival was dependent from amount of ADSC in the fat graft.
Autologous fat graft assisted by stromal vascular fraction improves facial skin quality: A randomized controlled trial [58]	A randomized control trial	CAL in 25 study group and fat only in 25 control group. The SVF cells were counted, tested in terms of viability, and characterized. The volumes of whole faces were determined by using a 3D scanner. Time of observation 12 months	CAL improves the outcomes and guarantees better and longer results.
Stem cell-enriched lipotransfer reverses the effects of fibrosis in systemic sclerosis [59]	Open cohort study	62 patients with scleroderma, injected with CAL, result evaluation with Mouth Handicap in Systemic Sclerosis Scale-MHISS, Cell viability, DNA content, protein secretion of known fibrotic mediators including growth factor- β1 (TGF β-1) and connective tissue growth factor (CTGF) using ELISA analysis	Fibrosis associated genes were down regulated: Matrix metalloproteinase-8 (MMMP-8), Platelet derived growth factor-β (PDGF-β) and Integrin Subunit Beta 6 (ITG-β6). CAL significantly improved the effects of oro-facial fibrosis.
Fat Grafting for Facial Rejuvenation Using Stromal Vascular Fraction Gel Injection [43]	Observative study	32 patients underwent transconjunctival eye bag removal with SVF-gel injection and 42 patients only received SVF-gel injection to correct tear trough deformity or infraorbital hollow.	High satisfaction was noted among patients treated with SVF-gel injection for periorbital rejuvenation with fairly low complication rates.
Composite Face Lifting: The Combination of Stromal Enriched Lipograft with Face Minilift and Upper and Lower Blepharoplasty: A Review of 210 Cases [60]	Prospective study	210 patients, evaluation after 6 months. Combination of minilift and upper and lower blepharoplasty with CAL. The amount of CAL transplanted varied from 22 to 56 mL per side (mean, 41)	Improvement of skin laxity, and skin quality that was synergic to surgery results. A safe method, no severe adverse events were reported.
Rejuvenation of facial skin and improvement in the dermal architecture by transplantation of autologous stromal vascular fraction: a clinical study [61]	Clinical study	16 patients, VF was harvested from 100 mL of harvested fat tissue, injected into NLF	Improvement in skin elasticity and density, improvement in thickness, as well as neovascularization were observed. No adverse events were reported.
Superficial Enhanced Fluid Fat Injection (SEFFI) to Correct Volume Defects and Skin Aging of the Face and Periocular Region [62]	Clinical study	98 patients with aesthetic procedures simultaneously with superficial liquid fat. minimal-incisions vertical endoscopic lift (n = 51), primary blepharoplasty (n = 35), neck lift (n = 23), and revisional blepharoplasty (n = 12), which were performed in a typical way. In second step during same procedure, the harvested and processed fat was injected in superficial layers with a 23G sharp needle	Volume restoration and improvement skin quality was observed. 3 minor side effects (cysts) reported.
The step-up approach in rejuvenation of the midface: a combination of minimally invasive procedures [63]	Clinical study	35 patients, 5 with simultaneous CAL and PDO barbed threads.	Simultaneous volumization, skin rejuvenation and tissue elevation were obtained. Prolonged edema up to 3 weeks was reported in 2 cases.
Introducing Platelet-Rich Stroma: Platelet-Rich Plasma (PRP) and Stromal Vascular Fraction (SVF) Combined for the Treatment of Androgenetic Alopecia [64]	Clinical study	10 male patients suffering from AGA at stage II to III, have been treated with a single injection of autologous PRS (ACPSVF: combination of PRP and SVF)	Hair density was significantly increased after 6 weeks and 12 weeks postinjection (*p* = 0.013 and *p* < 0.001). In hair-to-hair matching analyses, new hair grew from active follicles. Furthermore, nonfunctioning hair follicles filled with hyperkeratotic plugs, up to today assumed incapable of forming new hair, proved to grow new hair. No side effects were noted after treatment.
Adipose Stromal Vascular Fraction Gel Grafting: A New Method for Tissue Volumization and Rejuvenation [65]	Retrospective single center study	127 patients after SVF gel and 78 after conventional lipostransfer. SVF-gel were harvested and examined histologically	Mild side effects (swelling) in 10% of cases. 77% patients were very satisfied or satisfied with the results.
Application of Adipose Stem Cell Glue in Facial and Breast Plastic [66]	Retrospective study	60 patients who underwent facial and breast fat transplantation. Follow-up 8 months.	The satisfaction of patients after CAL was higher than after macrofat.
Nanofat grafting: basic research and clinical applications [67]	Clinical study	Nanofat grafting was performed in 67 cases to correct superficial rhytides, scars, and dark lower eyelids. In the research study, three fat samples (macrofat, microfat, nanofat) were analyzed.	No viable adipocytes were observed in the nanofat sample. Adipose-derived stem cells were still richly present in the nanofat sample. Cell cultures showed an equal proliferation and differentiation capacity of the stem cells from the three samples. Clinical applications showed remarkable improvements in skin quality 6 months postoperatively. No infections, fat cysts, granulomas, or other unwanted side effects were observed.
Changes in phenotype and differentiation potential of human mesenchymal stem cells aging in vitro [68]	Experimental	Human bone marrow-derived MSCs were passaged in vitro and cultivated, the gene expression profile and adipogenic and osteogenic was evaluated.	In vitro aging MSCs gradually lost the typical fibroblast-like shape, CD146 expression decreased. Stem cells undergo senescent changes.
Fat grafting for treatment of burns, burn scars, and other difficult wounds [69]	Observation study	240 patients with burns and chronic wound were treated with lipofilling with SVF.	Potential complications of the procedure are infection, edema, vessel injury, ulcerations. When used into scars it acts antifibrotic.
Fat Grafting and Adipose-Derived Regenerative Cells in Burn Wound Healing and Scarring: A Systematic Review of the Literature [70]	Systematic review	6 murine, 12 human studies	Lack of big, randomized studies, subjective improvement in scar appearance.
The effect of conditioned media of adipose-derived stem cells on wound healing after ablative fractional carbon dioxide laser resurfacing [71]	Prospective study	19 patients, CO2 laser, ADSC- CM applied topically over the wounds after CO2 resurfacing. These were cultured, human ADSC harvested from two liposuctions.	Application of ADSC improved healing.
Randomized controlled study for the anti-aging effect of human adipocyte-derived mesenchymal stem cell media combined with niacinamide after laser therapy [72]	Prospective study	25 patients, CO2 laser followed by topical application of DSC on a niacinamide vehicle.	Improvement in healing and wrinkles.
Nanofat Needling: A Novel Method for Uniform Delivery of Adipose-Derived Stromal Vascular Fraction into the Skin [73]	Clinical study	Topical application of nanofat with microneedling.	Improvement in skin quality and patient satisfaction was observed.

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
