# Peer review of "Adipose-Derived Stem Cells for Facial Rejuvenation"

_jpm, 2022, doi:10.3390/jpm12010117_

Round 1

Reviewer 1 Report

Comments to Author:

I read with interest “Adipose-derived stem cells for facial rejuvenation”.

The advantages and disadvantages of adipose derived stem cells have been well documented. This article report the powerful action of adipose derived stem cells in facial rejuvenation. Although the topic of ADSCs is very extensive, it is of particular importance, given the recent published papers that have raised interest in the use of ADSCs in different fields of regenerative medicine.

Considerations:

- I think you should report tables for all the articles have found in the literature about each sections (harvesting, powerful action, clinical indications, complications, future trends), for a better comprehension of the paper

- If the topic of the article was adipose derived stem cells and is useful in facial rejuvanation, why did you reported “a wide range of autologous products (i.ePRP)”? Did you compared it with ADSCs?

- In “Clinical indications for SVF” you talked about the use of cell assisted lipotransfer simultaneously with other surgical procedure. I think is important to explain the preparation of SVF for each surgical technique that you mentioned (facelift, transconjunctival lower blepharoplasty, threads)? From a practical point of view, I think is important to explain more in detail some literature experiences with the use of SVF for facial rejuvenation

- Could you advice the use of other techniques in combination with ADSCs for facial rejuvenation (i.e laser, radiofrequency radiation, etc)?

- At the end of complication you reported: “On the other hand, there are studies showing local progression of ductal adenocarcinoma in the breast and accelerated formation of distant metastases”. Klinger M et al (Safety of autologous fat grafting in breast cancer: a multicenter Italian study among 17 senonetwork breast units autologous fat grafting safety: a multicenter Italian retrospective study. Breast Cancer Res Treat; 2021) consolidated previous data on autologous fat grafting safety; their study confirmed in a very large, multicenter cohort of early breast cancer patients that, aside the well-known benefts on the esthetic result, adipose fat grafting do not interfere negatively with cancer prognosis. I think you should report it.

- Finally, you need to be very clear about what this adds to the existing literature and clearly detail learning points.

I believe the paper needs major revisions before it is accepted in Journal of Personalized Medicine.

Author Response

Dear Professor,

Thank you for the review. 

Thank you very much for all of your excellent remarks and considering my paper for publication in MDPI journal.

Please find below my answers to your comments and enclosed the revised manuscript.

According to your advice, I uploaded the Tables and summarized the cited articles.

PRP is often used in combination with SVF. Moreover, there are some false commercial data that peripheral blood may contain stem cells. In these paragraph I wanted to stress, that PRP is not a source of stem cells, however it has a rejuvenating power. If you suggest, this part could be removed.

There are not many works on simultaneous usage of SVF and surgeries, although a lot of surgeon perform such procedures. Thank you for suggestion, I am planning to describe and publish some of our protocols. I could not cite them as they are not in press yet. Same situation is with lasers and SVF. I perform CO2 with SVF but my results on scars have been only shown during AMWC but not full article has been published. 

Thank you for the suggestion regarding cancer. Klinger et al. transplanted "pure" fat in breast cancer, however I found an interesting study with enriched fat in G1 breast cancer patients.

I hope that you will be satisfied with the modifications of the manuscript.

Reviewer 2 Report

This is an interesting manuscript. The authors reviewed the adipose derived stem cells from the basic studies to clinical uses. This article is informative. However, I found somethings needed to be improved.

  1.  Page 4 line 11: Subsequently, the oil from the destroyed adipocytes on top and the aqueous fraction, containing the aesthetic and blood cells, are removed.... Please clarify the word " aesthetic " in this sentence.
  2. reference 28 is the Instruction for use : Puregraft 850 system. It is not an academic article. Please cites its original articles.
  3. There are several inconsistent format in the reference section. In reference 30~34,36,38,55...77,78, their line spacing and the font are different with others. Please revise them.

I suggest minor revision

Author Response

Dear Profesor,

Thank you for all the remarks. The manuscript has been modified according to Yours' and other Reviewer suggestions.

Regards,

This manuscript is a resubmission of an earlier submission. The following is a list of the peer review reports and author responses from that submission.